# Temperature–Humidity-Dependent Wind Effects on Physiological Heat Strain of Moderately Exercising Individuals Reproduced by the Universal Thermal Climate Index (UTCI)

**DOI:** 10.3390/biology12060802

**Published:** 2023-05-31

**Authors:** Peter Bröde, Bernhard Kampmann

**Affiliations:** 1Leibniz Research Centre for Working Environment and Human Factors at TU Dortmund (IfADo), Ardeystrasse 67, D-44139 Dortmund, Germany; 2Department of Occupational Health Science, School of Mechanical Engineering and Safety Engineering, University of Wuppertal, Gaußstraße 20, D-42119 Wuppertal, Germany; kampmann@uni-wuppertal.de

**Keywords:** heat stress, index, air temperature, humidity, wind, heat wave, electrical fan

## Abstract

**Simple Summary:**

With climate change exacerbating heat extremes, there is a growing need for sustainable measures that can reduce physiological heat strain. However, public health policies have issued warnings against the use of electric fans and ventilators at ambient temperatures exceeding typical skin temperatures of 35 °C. Aiming to extend previous findings for sedentary persons to moderately exercising individuals, we analyzed the heart rates, sweat rates, and core and skin temperatures recorded in 198 climatic chamber experiments with 3h exposures to treadmill work under widely varying heat stress conditions. Our findings suggest mitigating wind effects at temperatures above 35 °C when humidity is elevated. Moreover, the close agreement of the observed effects with the corresponding wind effects predicted by the Universal Thermal Climate Index (*UTCI)* demonstrates the potential of the *UTCI* for evaluating sustainable strategies of heat stress mitigation for moderately exercising individuals.

**Abstract:**

Increasing wind speed alleviates physiological heat strain; however, health policies have advised against using ventilators or fans under heat wave conditions with air temperatures above the typical skin temperature of 35 °C. Recent research, mostly with sedentary participants, suggests mitigating the effects of wind at even higher temperatures, depending on the humidity level. Our study aimed at exploring and quantifying whether such results are transferable to moderate exercise levels, and whether the Universal Thermal Climate Index (*UTCI*) reproduces those effects. We measured heart rates, core and skin temperatures, and sweat rates in 198 laboratory experiments completed by five young, semi-nude, heat-acclimated, moderately exercising males walking the treadmill at 4 km/h on the level for three hours under widely varying temperature–humidity combinations and two wind conditions. We quantified the cooling effect of increasing the wind speed from 0.3 to 2 m/s by fitting generalized additive models predicting the physiological heat stress responses depending on ambient temperature, humidity, and wind speed. We then compared the observed wind effects to the assessment performed by the *UTCI*. Increasing the wind speed lowered the physiological heat strain for air temperatures below 35 °C, but also for higher temperatures with humidity levels above 2 kPa water vapor pressure concerning heart rate and core temperature, and 3 kPa concerning skin temperature and sweat rate, respectively. The *UTCI* assessment of wind effects correlated positively with the observed changes in physiological responses, showing the closest agreement (r = 0.9) for skin temperature and sweat rate, where wind is known for elevating the relevant convective and evaporative heat transfer. These results demonstrate the potential of the *UTCI* for adequately assessing sustainable strategies for heat stress mitigation involving fans or ventilators, depending on temperature and humidity, for moderately exercising individuals.

## 1. Introduction

### 1.1. Sustainable Heat Stress Mitigation by Wind

Climate change accelerating with fossil energy use will exacerbate heat extremes, which will provoke physiological heat strain in terms of increased heart rates, core and skin temperatures, and sweat rates, imposing health risks [1,2], and potentially affecting human performance and productivity [3,4]. Elevating air movements by outdoor wind or ventilators is a cost-effective and sustainable measure for mitigating the physiological heat strain that is predominantly considered indoors and at stationary workplaces [5,6,7]. However, policies and public health authorities have advised against electric fan use under heat wave conditions indoors with air temperatures above the typical skin temperature of 35 °C, because convective cooling will then turn to convective heating of the body [8,9,10]. On the other hand, the well-known enhancement of sweat evaporation with increased wind speed [11,12,13] challenged these postulations and advocated for an evidence-based approach [14]. Recent studies, mostly with sedentary participants [15,16], have demonstrated the cooling effects of ventilation due to enhanced sweat evaporation at even higher temperatures at elevated humidity levels.

Comparative studies involving experimental heat exposures are usually restricted to a few well-defined climatic conditions [15]. Thus, for separating beneficial cooling from detrimental heating wind effects over a grid of temperature–humidity conditions in heat wave scenarios, several studies applied biophysical modeling based on human heat balance calculation [17,18,19,20]. Interestingly, qualitatively similar threshold curves based on modeling the naturally aspirated wet-bulb temperature were developed one century ago [21]. These curves indicated wind cooling at elevated humidity levels but additional heat loads in hot–arid climates, as flagged by the term ‘poison wind’ (*simoom*) for potentially fatal outdoor conditions in Southwest Asian desert regions [21]. Notably, the simulations of heat wave scenarios focused on indoor settings and only considered resting conditions concerning metabolic rates of about 1 MET (1 MET = 58.2 W/m^2^), whereas physical loads at workplaces and many home and outdoor activities are associated with metabolic rates higher than 2 MET [22,23].

Concerning physiological heat strain under moderate exercise levels, the recorded heart rates, core and skin temperatures, and sweat rates in Figure 1 from an extensive database of climate chamber experiments [24,25] exemplify the aforementioned findings for an acclimated young male walking at 4 km/h on a treadmill, corresponding to a 2.3 MET metabolic rate. While increasing the air velocity from 0.3 to 2 m/s lowered the physiological heat strain in a hot–humid climate with air temperature exceeding skin temperature (Figure 1a), increased wind amplified the heat strain under hot–dry conditions (Figure 1b). Figure A1 in Appendix A includes a supporting example from another participant. In both cases, the transient overshooting of the sweat rate with low wind (*v_a_* = 0.3 m/s) under warm–humid conditions (Figure 1a) may be explained by hidromeiosis [26], which did not occur when increasing the wind speed to *v_a_* = 2.0 m/s enhanced sweat evaporation [27]. On the other hand, under hot–dry conditions (Figure 1b), the already high level of evaporative efficiency could not be further elevated by increased airflow and thus could not compensate for the higher convective heat gain aggravating physiological strain, in particular rising sweat rates [28].

A recent study with exercising participants [28] examined how wind speed at various temperature–humidity combinations affected physical work capacity, defined by the workload related to metabolic rate, which the participants could tolerate with their heart rate clamped at 130 beats per minute (bpm). Coupling their experimental data with the heat balance calculations, they showed beneficial wind effects for air temperatures up to 44 °C with relative humidity exceeding 50% but detrimental wind effects for higher temperatures. Similar findings have been reported concerning the shift of the critical, i.e., tolerable, humidity level by wind in hot environments [29]. Another study [30] showed beneficial wind effects for sports activities at specific conditions at 37 °C air temperature and 50% relative humidity. 

In spite of the aforementioned research, and although Figure 1 and Figure A1 exhibit motivating examples, more systematic studies are required for quantification and evidence-based conclusions. Specifically, with a focus on moderately active persons, there is a lack of data from controlled laboratory experiments specifying wind effects on physiological heat strain in terms of heart rates, core and skin temperatures, and sweat rates while covering a wide grid of air temperature–humidity combinations, which determine the indoor thermal environment.

In addition to data requirements, there seems to be a need for improving the thermo-physiological modeling beyond the heat balance approach, as recent findings suggest that the simple models will underestimate physiological heat strain under electrical fan use, e.g., for the vulnerable elderly population [31].

When looking for advanced, but easy-to-use modeling approaches, the Universal Thermal Climate Index (*UTCI*) [32], developed for moderately active persons (2.3 MET), constitutes a noteworthy alternative. The index was constructed from the dynamic physiological thermal strain simulated by the advanced *UTCI*-Fiala model of human thermoregulation [33] coupled with an adaptive clothing model [34]. The *UTCI* allows for the assessment of the thermal environment, covering the range from extreme cold to extreme heat stress conditions. The model was extensively validated against laboratory data and field observations [35] and showed good agreement with thermal environment standards and experimental data in occupational settings [36,37,38] concerning the moderating influence of humidity, wind, and radiant heat load in addition to air temperature on both cold and heat stress. Though the *UTCI* relies on sophisticated models, the operational procedure [39] provides algorithms, making it easily applicable, e.g., for assessing the wind-cooling potential of actual and future urban scenarios in relation to wind direction [40]. Concerning the effects on physical work capacity, the *UTCI* outperformed any other considered thermal index in capturing not only the effects of wind over a wide grid of temperature–humidity conditions [28], but also in combination with heat radiation [41].

### 1.2. Study Objectives

Based on a comprehensive physiological heat strain database [24], this study aimed to explore and quantify whether reports on beneficial wind effects with air temperature exceeding skin temperature under humid conditions for sedentary individuals are transferable to moderate exercise levels, and whether the Universal Thermal Climate Index (*UTCI*) reproduces these effects.

Our approach was to determine the transition from cooling to heating wind effects on heart rates, sweat rates, and core and skin temperatures measured in experiments with exercising participants over a comprehensive matrix of temperature–humidity combinations and to compare the outcomes to the UTCI assessment.

## 2. Materials and Methods

### 2.1. Experimental Data

This experimental study is based on a database compiled from about 2400 climate chamber experiments with 33 young male adults, conducted previously at IfADo [24,42], under widely varying climatic conditions, clothing, and activity levels. The experiments were carried out according to the ethical principles of the Declaration of Helsinki after approval by IfADo’s local Ethics Committee.

We searched our database for series of heat stress experiments with participants exercising on a treadmill under both reference (***RefWind***, with air velocity *v_a_* = 0.3 m/s) and high wind (***HiWind***, *v_a_* = 2 m/s) conditions. Here, 2 m/s constituted the maximum value of controllable wind speed in the climatic chamber, corresponding to average airflow conditions observable in (semi-)outdoor workplaces and in coal mines [24]. Inclusion criteria were a minimum number of 15 experiments per series with comparable workloads and clothing. We retrieved 198 trials organized in 10 series, which originated from five acclimated, semi-nude (basic clothing insulation *I_cl_* = 0.1 clo, 1 clo = 0.155 K·m^2^·W^−1^), young, and fit males under *RefWind* and *HiWind* conditions, respectively. The number of experiments in each series varied inter-individually and depended on wind conditions between 16 and 25 experiments, with total numbers of 97 trials for *RefWind* and 101 for *HiWind*. The average personal characteristics (mean ± SD, with range in brackets) of the five participants were 20.1 ± 0.9 (19–22) years of age, 1.87 ± 0.02 (1.84–1.88) m of body height, 70.5 ± 2.1 (68–73) kg of body weight, 1.94 ± 0.02 (1.92–1.97) m^2^ of body surface area, and 47.9 ± 6.4 (43.2–57.4) mL/min/kg of maximal oxygen consumption. Before exposure, the participants had undergone a heat acclimation protocol lasting at least three weeks [43].

We will briefly summarize the procedures as detailed descriptions are available elsewhere [24]. Each trial consisted of treadmill work with constant workload of walking 4 km/h on the level, which corresponded to the activity level of 2.3 MET, as assumed for *UTCI* [39]. Each trial lasted for at least three hours and consisted of 30 min work periods interrupted by 3 min breaks for determining body weight loss, from which sweat rate (*SR*) was calculated correcting for ad libitum fluid ingestion. Under both *HiWind* and *RefWind*, the participants were exposed to varying levels of heat stress, with conditions characterized by different combinations of air temperature (*T_a_*; range 25–55 °C) and humidity, expressed as water vapor pressure (*p_a_*; 0.5–5.3 kPa). Mean radiant temperature (*T_mrt_*) was equal to *T_a_*. Trials were stopped prematurely if rectal temperature exceeded 38.5 °C or on the participant’s demand.

Rectal temperatures (*T_re_*) were recorded continuously using a thermistor probe (YSI 401, YSI Inc., Yellow Springs, OH, USA) inserted 10 cm past the anal sphincter. Local skin temperatures were measured with thermistors (YSI 427, YSI Inc., Yellow Springs, OH, USA) at the forehead (*T_sk*,*head_*), chest (*T_sk*,*chest_*), back (*T_sk*,*back_*), upper arm (*T_sk*,*arm_*), thigh (*T_sk*,*thigh_*), and lower leg (*T_sk*,*leg_*), and were used to calculate mean skin temperature (*T_sk_*) as weighted average of the local skin temperatures according to [24], shown in Equation (1):*T_sk_* = 0.05 × *T_sk*,*head_* + 0.2 × *T_sk*,*chest_* + 0.15 × *T_sk*,*back_* + 0.2 × *T_sk*,*arm_* + 0.25 × *T_sk*,*thigh_* + 0.15 × *T_sk*,*leg_*(1)

Heart rates (*HR*) were obtained using ECG electrodes and stored in one-minute intervals, as were *T_re_* and *T_sk_*. As illustrated by the yellow shaded areas in Figure 1, the averages of *T_re_*, *HR*, *T_sk_*, and *SR* over the third hour of exposure, representing steady-state [25], were submitted to the following data analysis.

### 2.2. Data Analysis and Statistics

Penalized regression splines fitted by generalized additive models (GAMs) provide a flexible modeling framework for irregularly distributed data [44], allowing for non-linear [45] and random effects [46].
*E*[*Y_ij_*]*
*=* µ *+* s*(*ID_i_*)*
*+* te*(*T_a*,*ij_*,* p_a*,*ij_*)*
*+* te_Δv_*(*T_a*,*ij_*,* p_a*,*ij_*)*
*+* ε_ij_*(2)

Using the notation from Equation (2) [44], we predicted *HR*, *T_re_*, *T_sk_*, and *SR*, respectively, as responses from experiment *j* of participant *i* (*Y_ij_*) by separate GAMs. These models included an overall intercept *µ*, and a bivariate penalized tensor regression spline *te*(*T_a*,*ij_*,* p_a*,*ij_*)** for the effects of air temperature and humidity depending on the (*T_a_*,* p_a_*) combinations, accounting for the repeated measurements by subject-specific intercepts as random coefficients *s*(*ID_i_*)** [46]. Adding another bivariate spline *te_Δv_*(*T_a*,*ij_*,* p_a*,*ij_*)** as so-called factor smooth interaction [44] of the regression splines with the wind condition (*RefWind* vs. *HiWind*), we obtained estimates for the wind effect *Δ_v_*, i.e., the response difference under *HiWind* compared to *RefWind* over the (*T_a_, p_a_*) grid supplemented by *p*-values [47]. These were then visualized in a difference plot [48].

More specifically, *s(ID_i_)* refers to a cubic regression spline with basis dimension (or rank) set to k = 5 (i.e., with k − 1 = 4 as upper limit of the associated degrees of freedom). Similarly, *te*(*T_a*,*ij_*,* p_a*,*ij_*)** and *te_Δv_*(*T_a*,*ij_*,* p_a*,*ij_*)** represent tensor products of two cubic regression spline bases for the *T_a_* and *p_a_* dimension, respectively, each of rank k = 9, and are hence of total rank k = 81. Maximum likelihood parameter estimates with standard errors and *p*-values were obtained assuming Gaussian error *ϵ_ij_* [47]. Calculations were performed with the R software version 4.2.1 [49] using the package *mgcv* [44] together with *mgcViz* [48] and *tidymv* [50].

### 2.3. UTCI Calculations

We calculated *UTCI* values using the regression polynomial, as described in the *UTCI* operational procedure [39] for given combinations of air temperature (*T_a_*), ambient water vapor pressure (*p_a_*), mean radiant temperature (*T_mrt_*), and air velocity 10 m above ground (*v_a*,10*m_*).

#### 2.3.1. UTCI Sensitivity to Wind

Following meteorological conventions, *UTCI* calculations rely on air velocity 10 m above ground (*v_a*,10*m_*). For conversion to any other measurement height, e.g., 1 m for person level, the operational procedure [39] provides a logarithmic formula, shown in Equation (3), indicating that air velocity at person level (*v_a*,1*m_*) is computed as the 10 m value (*v_a*,10*m_*) divided by 1.5, in accordance with international standards [51].
*v_a*,1*m_* = *v_a*,10*m_
*×* log*(1/0.01)*/log*(10/0.01) **=* v_a*,10*m_*/1.5(3)

As the *UTCI* person is assumed to move on the level at 4 km/h, corresponding to a walking speed of *v_w_* = 1.1 m/s, this is taken into account by calculating the resulting or relative air velocity at person level (*v*_*ar*,1*m*_), according to Equation (4). Here, α denotes the angle between the directions of walking and wind assigned to zero for indicating the same direction. As *UTCI* does not assume a specific angle, *v*_*ar*,1*m*_ is calculated by integrating Equation (4) over all α between zero and 2π [34,52].
(4)$$v_{ar,1m}=\sqrt{{\left(v_{w}{-v}_{a,1m}\times cos\alpha \right)}^{2}+{\left(v_{a,1m}\times sin\alpha \right)}^{2}}$$

Figure 2 presents the resulting *v*_*ar*,1*m*_ used by *UTCI* for calculating the convective and evaporative heat loss [33]. Notably, reducing wind speed below *v_a*,10*m_* = 0.5 m/s (*v_a*,1*m_* = 0.3 m/s) will hardly impact *v*_*ar*,1*m*_, which is limited by *v_w_*, while *v*_*ar*,1*m*_ will approach *v_a*,1*m_* for *v_a*,10*m_* above 3 m/s.

#### 2.3.2. Wind Effect on UTCI

We computed the effect of actual wind speed *v_a*,10*m_* on *UTCI* (*Δ_v_UTCI*) compared to the reference wind speed (*v_a*,10*m*,*ref_* = 0.5 m/s) while keeping the other parameters constant, as shown in Equation (5):*∆_v_UTCI *=* UTCI*(*T_a_*;*p_a_*;*v_a*,10*m_*;*T_mrt_*)*
*−* UTCI*(*T_a_*;*p_a_*;*v_a*,10*m*,*ref_*;*T_mrt_*)**(5)

Matching the range of experimental climatic conditions (Section 2.1), but respecting the upper limits of *UTCI* validity [39,53], we computed *UTCI* and *Δ_v_UTCI* for combinations of *T_a_* between 25 and 50 °C and of *p_a_* from 0.1 to 5 kPa with relative humidity of *rH* ≤ 100% while setting *T_mrt_ = T_a_*, as in the experiments. We performed the calculations for the reference wind speed (*v_a*,10*m*,*ref_* = 0.5 m/s), matching *RefWind* with *v_a*,1*m_* = 0.3 m/s (Figure 2), and increased wind speeds with *v_a*,10*m_* = 3 m/s, matching *HiWind* with *v_a*,1*m_* = 2 m/s.

Supplemental calculations were performed for assessing the modifying effect of radiant heat load by increasing Δ*T_mrt_ = T_mrt_−T_a_* from 0 to 30 K in steps of 10 K. In addition, we considered conditions with higher wind speeds of *v_a*,10*m_* = 4 and 6 m/s, respectively. Here, *v_a*,10*m_* = 4 m/s corresponds to an increase of 1.7 m/s in relative air velocity at body level (*v*_*ar*,1*m*_), according to Figure 2, matching the settings of the climate chamber experiments, whereas *v_a*,10*m_* = 6 m/s (*v_a*,1*m_* = 4 m/s) parallels the wind speed applied in several simulations and experimental studies [15,17,18,19,20,28].

#### 2.3.3. UTCI Prediction of Physiological Wind Effects

We compared the *UTCI* assessment of the wind effects *Δ_v_UTCI* with the wind effects obtained for the physiological variables by scatterplots, including Spearman correlation coefficients (*r_s_*). In addition, we quantified the predictive performance of *UTCI* in the binary classification of the occasion of heating wind effects (*Δ_v_* > 0) on physiological variables by calculating the sensitivity, specificity, and overall accuracy.

## 3. Results

In order to illustrate the structure of our data analysis approach, and deviating from the order of topics in the methods section with the experimental study preceding *UTCI* calculations, we first present in Section 3.1 the wind effects (*HiWind* vs. *RefWind*) evaluated by the *UTCI*. Section 3.2 contains the corresponding analyses for the experimental heat strain data, while Section 3.3 relates the physiological wind effects to the *UTCI* assessment. Eventually, Section 3.4 shows the results for the supplemental *UTCI* analyses considering higher wind speeds and the influence of thermal radiation.

### 3.1. Wind Effects Assessed by UTCI

The psychrometric charts in Figure 3 display the *UTCI* contours for selected index values, representing the upper limits of the *UTCI* heat stress categories in relation to the air temperature and humidity for the different wind speeds (Figure 3a). The contours were bended upward to the left, indicating that the heat stress increased with temperature and humidity, as expected [36,39]. The dashed lines (*HiWind*) were above the solid lines (*RefWind*), indicating wind cooling at temperatures below 34–35 °C and in hot–humid conditions. On the other hand, dashed lines below solid lines indicating heating by wind were found in hot–dry conditions. The dot-dashed line connecting the intersections of the *RefWind* and *HiWind* contours represents the zero wind effect *Δ_v_UTCI* = 0.

Figure 3b quantifies the magnitude of the wind effect *Δ_v_UTCI* depending on the temperature and humidity. Wind cooling (blueish) increased with a decreasing temperature and increasing humidity, while heating due to wind (reddish) increased with an increasing temperature and decreasing humidity.

### 3.2. Wind Effects on Physiological Heat Strain

Table 1 summarizes the results of the GAMs fitted separately to the physiological heat strain variables. While the data for *T_re_* were complete, there were a few missing values for *HR*, *T_sk_*, and *SR*. The estimated mean values of 103 bpm (*HR*), 37.6 °C (*T_re_*), 35.5 °C (*T_sk_*), and 744 g/h (*SR*), respectively, were typical for light to moderate activities under heat stress [24,38]. The goodness-of-fit was good to excellent, with more than three quarters of the variance explained (*R*^2^ > 75%) and small residual standard errors for *HR*, *T_re_*, and *T_sk_*. *SR* showed almost a 90% explained variance but a slightly increased standard error, which, however, was still below 125 g/h, considered the relevant accuracy limit acceptable in occupational or military settings [54]. Moreover, graphical model checking, as displayed in Appendix A in Figure A2, did not reveal any problematic issues concerning the underlying model assumptions.

For all heat strain variables, the splines *te(T_a_, p_a_)* modeling the influence of temperature and humidity were statistically highly significant (*p* < 0.0001). Similarly, the moderating effects of wind speed, as considered by the bivariate interaction splines *te_Δv_(T_a_, p_a_)*, were statistically significant for *HR*, *T_re_*, *T_sk_*, and *SR*.

Inter-individual variability, as indicated by the random term *s(ID)*, played a considerable role (*p* < 0.0001) for *T_re_*, *T_sk_*, and *SR*, respectively, whereas individual influences on *HR* appeared to be less pronounced (*p* = 0.20). 

Analogous to the *UTCI* results from Figure 3a, Figure 4 displays the effect of temperature and humidity separately for *RefWind* and *HiWind*, as contour lines for the physiological heat strain responses of *HR*, *T_re_*, *T_sk_*, and *SR*. Although the shape of the lines varied between the four variables and additionally depended on the strain level, a general pattern emerged, with contours bended upward to the left indicating strain levels increasing with temperature and humidity, as had also occurred for *UTCI* (Figure 3a). Again, similar to the *UTCI*, the dashed lines (*HiWind*) were above the solid lines (*RefWind*) at low temperatures and under hot–humid conditions, demonstrating reduced strain due to wind cooling, whereas in hot–dry climates, the opposite was found, with solid lines above dashed lines indicating higher strain levels with increased wind speed. 

The dot-dashed lines represent the zero wind effect *Δ_v_* = 0, thus demonstrating that the threshold separating cooling from heating effects was bended upward to the right, resemblingthe *UTCI* (Figure 3b), but with a greater curvature. They indicated reduced heat strain with *HiWind* at low temperatures and in hot–humid conditions, with limiting vapor pressure above 2 kPa for *HR* and *T_re_* and above 3 kPa for *T_sk_* and *SR*, respectively.

The difference plots in Figure 5 show the intensity of the cooling (blueish) and heating (reddish) wind effects over the temperature–humidity grid estimated by GAM, where white areas indicate indifferent temperature–humidity combinations with statistically non-significant differences from *Δ_v_* = 0.

Whereas no significant core temperature increase due to wind was observed, detrimental wind effects with increased *HR*, *T_sk_*, and *SR* were found under hot–arid conditions with a temperature above 40 °C and vapor pressure below 1 kPa. Temperatures below 35 °C yielded no detrimental wind effects.

Though significant decreases in the core temperature and heart rates by wind under high temperatures started at vapor pressures above 2 kPa, pronounced effects with reductions of more than 10 bpm *HR* or 0.15 °C *T_re_* only appeared for very humid conditions with vapor pressures approaching 4 kPa.

### 3.3. UTCI Assessment Related to Physiological Wind Effects 

The scatterplots in Figure 6a comparing the wind effects (*Δ_v_*) on the *UTCI* with the physiological effects show positive correlations with the closest agreement (*r* ≈ 0.9) for skin temperature and sweat rate, also concerning the transition from cooling (*Δ_v_* < 0) to heating (*Δ_v_* > 0) effects. Slightly lower correlations were observed with the effects on heart rates and core temperature.

In addition to correlation analyses, the plots give an indication of the capacity of the *UTCI* for correctly classifying conditions with physiological wind cooling or heating [56]. The dashed reference lines in Figure 6a divide the plot into quadrants, with the lower left quadrant showing the physiological cooling events correctly assessed by the *UTCI* and the upper right containing the correctly assessed heating events, which dominate concerning *T_sk_* and *SR*.

Whereas hardly any data points were present in the upper left quadrant with wind cooling falsely predicted by the *UTCI*, the conditions collected in the lower right quadrants (falsely predicted heating) concur with a more conservative *UTCI* assessment of the transition from cooling to heating wind concerning *HR* and *T_re_*.

When quantifying the *UTCI* performance for predicting the occasion of heating wind effects (*Δ_v_* > 0) as sensitivity, specificity, and accuracy in Figure 6b, these falsely predicted heating events resulted in lowered specificity and overall accuracy regarding *HR* and *T_re_*, whereas well-balanced figures emerged for *T_sk_* and *SR*.

### 3.4. UTCI Assessment with Higher Wind Speeds and Thermal Radiation

Figure A3 in Appendix A visualizes the wind effects (*Δ_v_*) on the *UTCI* over the temperature–humidity grid for the different wind speeds and radiant heat loads, expressed as *ΔT_mrt_ = T_mrt_* – *T_a_*, with the upper left panel with *v_a*,10*m_* = 3 m/s and *ΔT_mrt_* = 0 K repeating the data from Figure 3b. Figure 7 summarizes the wind effects *Δ_v_UTCI* depending on the wind speed and radiant heat load as boxplots calculated over the temperature–humidity grid.

As expected, increasing the wind speed amplified both the cooling (*Δ_v_UTCI* < 0) and heating effects (*Δ_v_UTCI* > 0) at the extremes, as indicated by the minima and maxima, but also the first and third quartiles in Figure 7. In addition, the threshold line for *Δ_v_UTCI* = 0 slightly shifted toward higher temperatures for higher wind speeds (Figure A3), thus contributing to the decreasing trend with wind speed observed for the median effect in Figure 7.

Concerning the influence of heat radiation on *Δ_v_UTCI*, all percentiles showed a decreasing trend with an increasing radiant heat load *ΔT_mrt_* (Figure 7). The reduced reddish surface areas with an increased radiant heat load in Figure A3 suggest that this was attributable to a massive shift in the threshold line toward higher temperatures. Thus, wind-cooling effects occurred for a higher portion of climatic conditions over the temperature–humidity grid, shifting the *Δ_v_UTCI* distribution toward more negative values (Figure 7).

The wind effect *Δ_v_UTCI* at *v_a*,10*m_* = 4 m/s (with *ΔT_mrt_* = 0 K) corresponded to the effect of increasing relative air velocity (*v*_*ar*,1*m*_) by 1.7 m/s (Figure 2), conforming to the change in the (relative) air velocity realized in the experiments. As the relevant *UTCI* calculations of convective and evaporative heat losses depend on the relative air velocity [33], we repeated the correlational analyses from Figure 6 for this condition, as shown in Figure A4. The outcome was very similar as before in Figure 6, though we observed slightly lowered correlations of the physiological wind effects with *Δ_v_UTCI*.

## 4. Discussion

This study achieved its primary aim to provide heat stress thresholds over a wide matrix of temperature–humidity combinations separating cooling from heating wind effects based on experimental evidence for moderately exercising individuals. 

### 4.1. Temperature-Humdity-Dependent Wind Effect Thresholds

With a view on sustainable heat stress mitigation for moderate activity levels occurring during domestic or industrial work, we assessed the wind effects on physiological heat strain parameters connected with the use of, e.g., electric fans or ventilators. Linking a comprehensive database originating from climate chamber experiments [24] with flexible spline regression models [44], we quantified the physiological heat strain responses of moderately exercising individuals to increasing wind speed over a wide range of temperature–humidity combinations. Thus, we could derive the temperature–humidity-dependent thresholds marking the transition from cooling (*Δ_v_* < 0) to heating (*Δ_v_* > 0) wind effects on physiological heat strain based on the experimental data. This is a distinguishing feature and particular strength of our study because corresponding experiments with sedentary participants have been limited to a few selected climatic conditions [15,16], so mapping the wind effect thresholds in relation to air temperature and humidity for low-active persons had to rely on simulation studies [17,18,19,20]. Recently, a study with exercising participants [28] aimed at the interaction of wind with combinations of temperature and humidity over a likewise extensive grid but focused on the physical work capacity with cardiac strain clamped at 130 bpm *HR*.

Our experimental findings suggest beneficial, i.e., strain-reducing wind effects for temperatures below 35 °C, conforming to common public health recommendations [8,9,10]. However, beneficial wind effects were also observed at higher temperatures with humidity levels above 2 kPa water vapor pressure concerning heart rate and core temperature, and 3 kPa concerning skin temperature and sweat rate, respectively (Figure 5). This agreed qualitatively with the corresponding charts produced in previous studies [19,28], though there were discrepancies concerning the humidity level separating beneficial wind cooling from detrimental heating effects, which was specified at 50% relative humidity in the former studies, while our study suggested thresholds in terms of absolute humidity (vapor pressure). Some of these discrepancies may be explained by the 3 h exposures in our study, whereas shorter exposures (1 h) had been used in previous studies [28], with longer exposure times usually resulting in higher strain levels [25,57]. The lower humidity limits for skin temperature and sweat rates compared to core temperature and heart rates in our study may reflect the direct influence of wind speed on the heat transfer from the skin surface by convection and evaporation and on sweating efficiency [11,12,13,58].

For the *UTCI*, no limiting humidity in terms of vapor pressure was identified, but wind-cooling effects were evident for any temperature condition with a relative humidity above 50% (Figure 3b). The *UTCI*’s assessment of wind effects showed high positive correlations with the observed changes in physiological responses (Figure 6a), exhibiting the closest agreement (*r* = 0.9) for skin temperature and sweat rate, where wind is known for elevating the relevant convective and evaporative heat transfer as discussed before. The *UTCI* also showed good agreement concerning the transition from cooling (*Δ_v_* < 0) to heating (*Δ_v_* > 0) wind effects on skin temperatures and sweat rates. The *UTCI*’s assessment was more conservative, i.e., overpredictive, concerning the potential hazards due to additional heating by wind with respect to core temperature and heart rates (Figure 6b). 

The expanded *UTCI* calculations suggested that further increasing wind speed would amplify the cooling and heating effects depending on temperature and humidity (Figure 7 and Figure A3). In addition, these calculations pointed to a relevant interaction with the radiant heat load, e.g., arising from outdoor exposure to solar radiation. The elevated radiant heat load will increase the *UTCI* by approximately 3 K per 10 K increase in *ΔT_mrt_* [36]. On the other hand, the limit of beneficial wind cooling was shifted toward higher temperatures, counteracting, at least partly, the heat gained from the increased radiant heat load. Similar interaction effects with wind decreasing the heat gain from radiant heat have been reported from thermal manikin measurements with work clothes [59,60]. However, as our results concerning the effects of heat radiation are purely modeled outcomes using UTCI, further supporting experimental studies with human participants concerning the interaction of wind with radiant heat load are required.

### 4.2. Limitations and Outlook

The selected study population of young, healthy, and fit males certainly limits the generalizability of our findings, though recent research suggests only minimal influences of individual factors on the heat stress assessment under moderate activity levels [61], as applied in our study. As another limitation, the responses of semi-nude participants (*I_cl_* = 0.1 clo) might differ from the effects with clothed humans, e.g., as assumed by the *UTCI* with clothing insulation not falling below *I_cl_* = 0.3 clo [34]. On the other hand, prior studies with sedentary participants [15,16] applied similar minimal clothing settings. In addition, limited differences were found concerning the wind effects on the physical work capacity with minimal clothing compared to light and covering clothing [28], and the corresponding temperature–humidity charts in that study were produced for a person wearing *I_cl_* = 0.28 clo.

Given these limitations, future experimental and modeling studies should consider the moderating influence of individual factors such as age, sex, or body constitution and of solar radiation and/or clothing with relevance, e.g., for desert conditions (*simoom*) [21], but also for (semi-)outdoor workplaces.

## 5. Conclusions

By widening the previous findings from resting persons to moderately exercising individuals, our results confirm the wind-cooling effects for air temperatures below 35 °C, and corroborate and quantify the potential wind-cooling effects at air temperatures above typical skin temperature values in humid climates.

In addition, our findings suggest that the *UTCI* has the capability to adequately assess the heat stress mitigation by wind depending on temperature and humidity under such conditions. Thus, the *UTCI* constitutes an easy-to-use tool that may be helpful for establishing evidence-based policies aiming at sustainable heat mitigation strategies under climate change scenarios.

## Figures and Tables

**Figure 1 biology-12-00802-f001:**
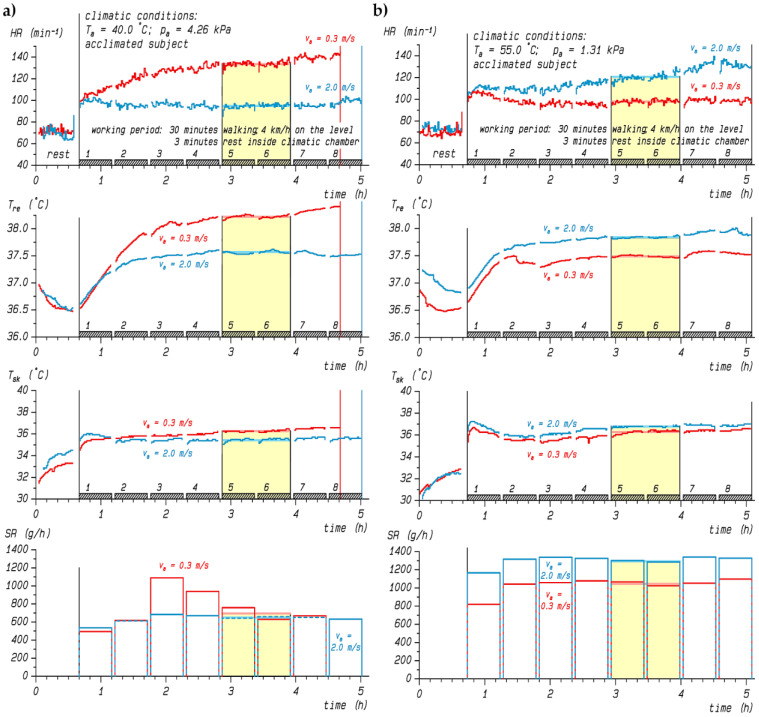
Time course of heart rate (*HR*), rectal (*T_re_*) and mean skin temperature (*T_sk_*), and sweat rate (*SR*) of an acclimated participant during experiments in a hot–humid climate (**a**) or in a hot–dry climate (**b**) under both reference (*v_a_* = 0.3 m/s, red lines) and high air velocity (*v_a_* = 2 m/s, blue lines) conditions. The yellow shaded area marks the time interval with averaged values used for analyses. Note that the reference wind trial in (**a**) was prematurely aborted in the 8th working period.

**Figure 2 biology-12-00802-f002:**
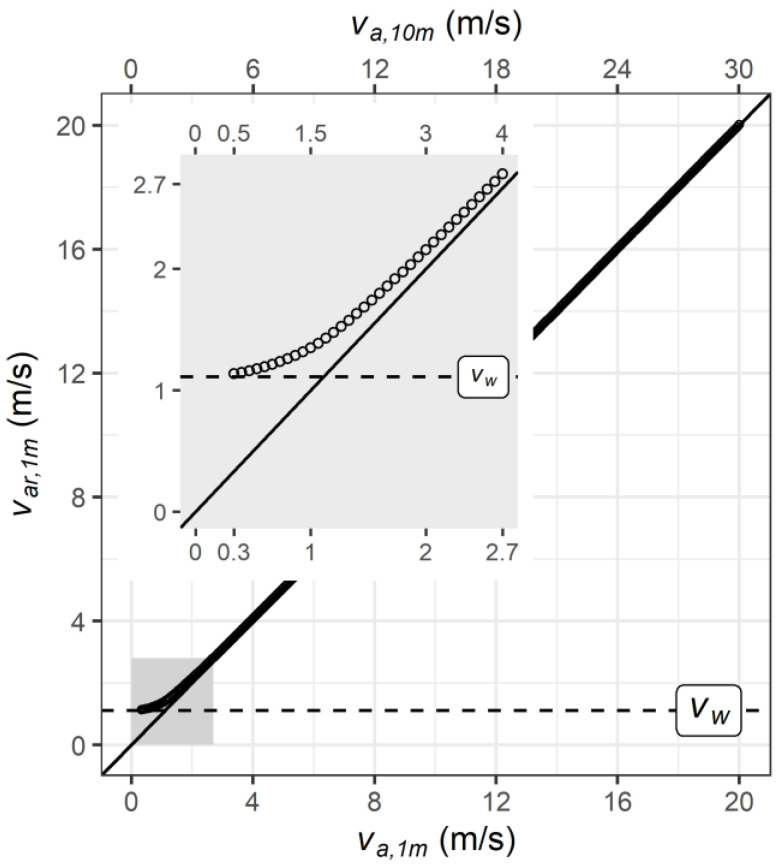
Relative wind speed at person level (*v*_*ar*,1*m*_) when moving with the *UTCI* reference walking speed (*v_w_*) of 4 km/h, corresponding to 1.1 m/s, in relation to wind speed measured at person level (*v_a*,1*m_*) and 10 m above ground (*v_a*,10*m_*). The insert provides a detailed view of the gray-shaded region for *v_a*,10*m_* ≤ 4 m/s. Figures include solid lines of identity, and horizontal dashed lines indicating walking speed (*v_w_*).

**Figure 3 biology-12-00802-f003:**
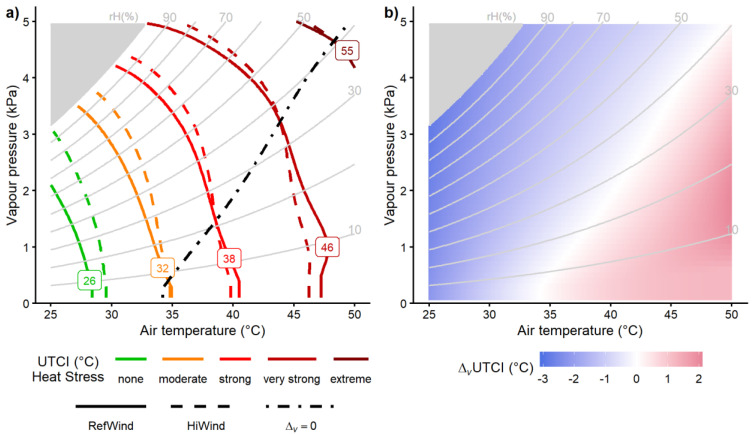
Psychrometric charts showing (**a**) *UTCI* contours related to air temperature and humidity (vapor pressure) representing the upper limits of ‘no thermal stress’ (*UTCI* = 26 °C), and ‘moderate’ (32 °C), ‘strong’ (38 °C), and ‘very strong’ (46 °C) heat stress, as well as an illustrative value for ‘extreme heat stress’ (55 °C), respectively. The influence of increased wind speed (*HiWind* with *v_a*,10*m_* = 3 m/s, corresponding to *v_a*,1*m_* = 2 m/s: dashed lines) is compared to the reference *RefWind* (*v_a*,10*m_* = 0.5 m/s, corresponding to *v_a*,1*m_* = 0.3 m/s: solid lines). The dot−dashed line connecting the intersections of *RefWind* with *HiWind* contours represents zero wind effects (*Δ_v_* = 0). This line is redrawn in (**b**) as white contour together with the colored regions, indicating temperature–humidity combinations with cooling (blueish) or heating (reddish) wind effects on *UTCI* (*Δ_v_UTCI*), according to Equation (5), respectively, where color intensity indicates magnitude. Gray contours mark relative humidity (*rH*) levels. All values were calculated without additional radiant heat load (*ΔT_mrt_* = *T_mrt_* − *T_a_* = 0 K).

**Figure 4 biology-12-00802-f004:**
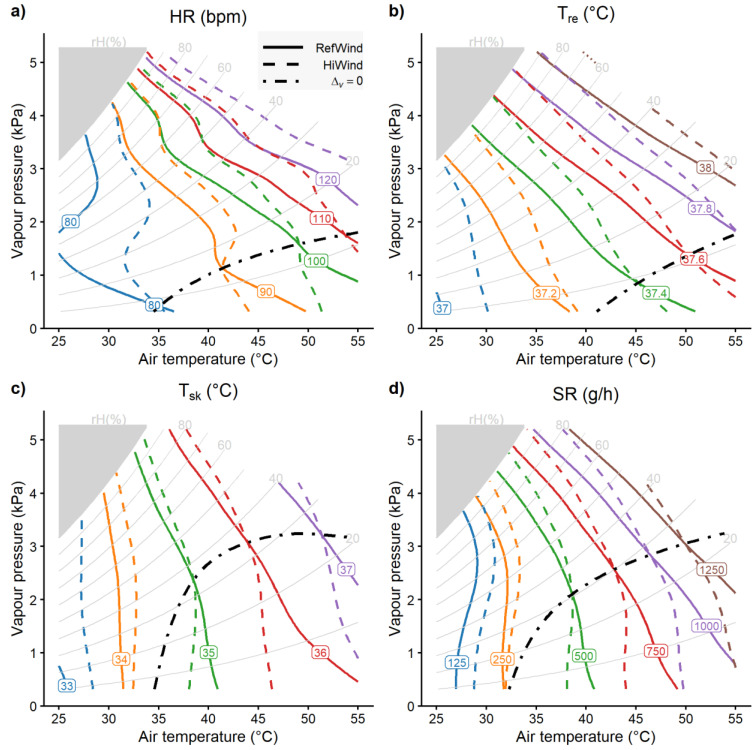
Psychrometric charts including contour lines with colors indicating separate levels of predicted values fitted by GAMs to (**a**) heart rates (*HR*), (**b**) rectal temperatures (*T_re_*), (**c**) mean skin temperatures (*T_sk_*), and (**d**) sweat rates (*SR*) in relation to air temperature and vapor pressure under *RefWind* (solid lines) and *HiWind* conditions (dashed lines), respectively. Dot-dashed lines connecting the intersections of *RefWind* with *HiWind* contours indicate zero wind effects (*Δ_v_* = 0).

**Figure 5 biology-12-00802-f005:**
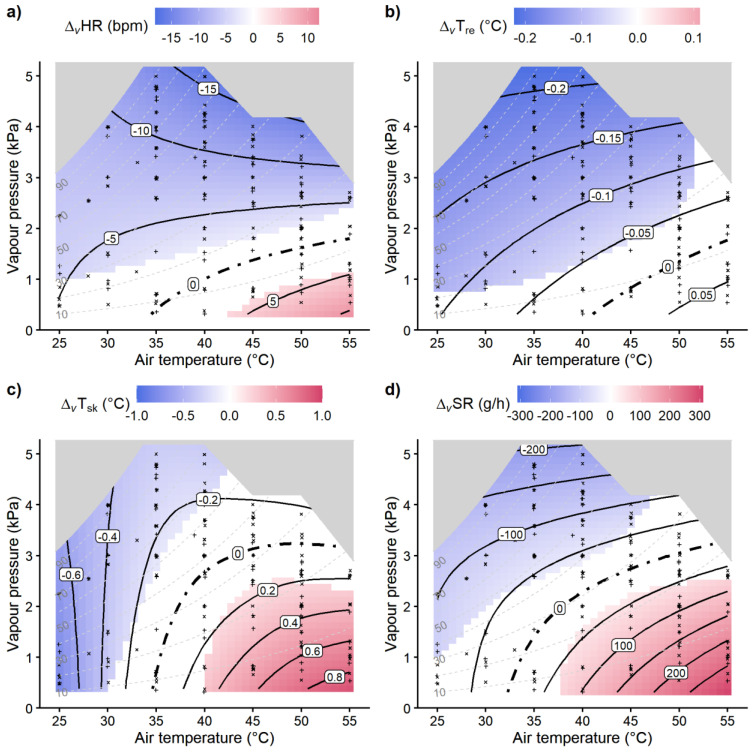
Difference plots with contours of the wind effects (*Δ_V_*) estimated by GAMs in relation to air temperature and vapor pressure for (**a**) heart rates (*HR*), (**b**) rectal temperatures (*T_re_*), (**c**) mean skin temperatures (*T_sk_*), and (**d**) sweat rates (*SR*). White areas indicate temperature−humidity regions with non−significant wind effects (*p* > 0.05 testing for *Δ_v_* = 0 vs. *Δ_v_* ≠ 0), the blueish areas mark significant reductions in physiological strain by wind cooling, while reddish areas depict significantly increased strain levels due to heating wind effects, respectively. Dashed lines show relative humidity levels, and the gray shaded areas indicate combinations not supported by data. Experimental conditions are marked by ‘+’ (*RefWind*) and ‘×’ (*HiWind*), respectively.

**Figure 6 biology-12-00802-f006:**
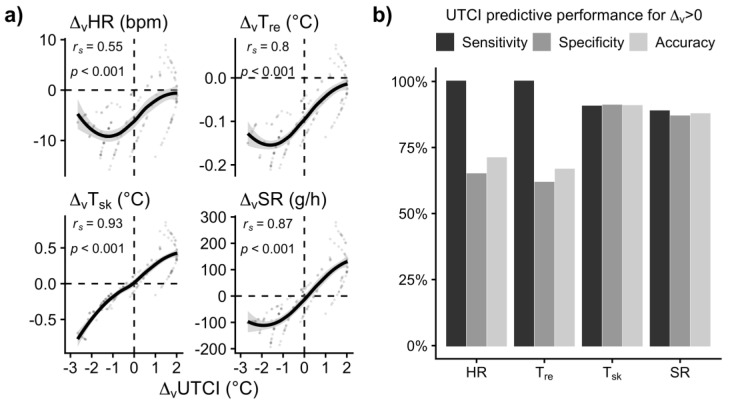
(**a**) Correlation of wind effects from Figure 5 for heart rate (*Δ_v_HR*), rectal temperature (*Δ_v_T_re_*), mean skin temperature (*Δ_v_T_sk_*), and sweat rate (*Δ_v_SR*), respectively, with the wind effect on *UTCI* (*Δ_v_UTCI*), shown in Figure 3b, calculated for the temperature–humidity combinations of the experiments (cf. Figure 5). Spearman correlations (*r_s_*) and *p*−values are shown with smoothing splines and 95% confidence bands. Dashed vertical and horizontal lines indicate zero wind effects (*Δ_v_* = 0) for *UTCI* and the physiological responses, respectively, where negative values (*Δ_v_* < 0) indicate wind cooling and positive values (*Δ_v_* > 0) heating effects. (**b**) UTCI performance in predicting heating wind effects (*Δ_v_* > 0) on *HR*, *T_re_*, *T_sk_*, and *SR*, respectively, expressed by sensitivity, specificity and accuracy.

**Figure 7 biology-12-00802-f007:**
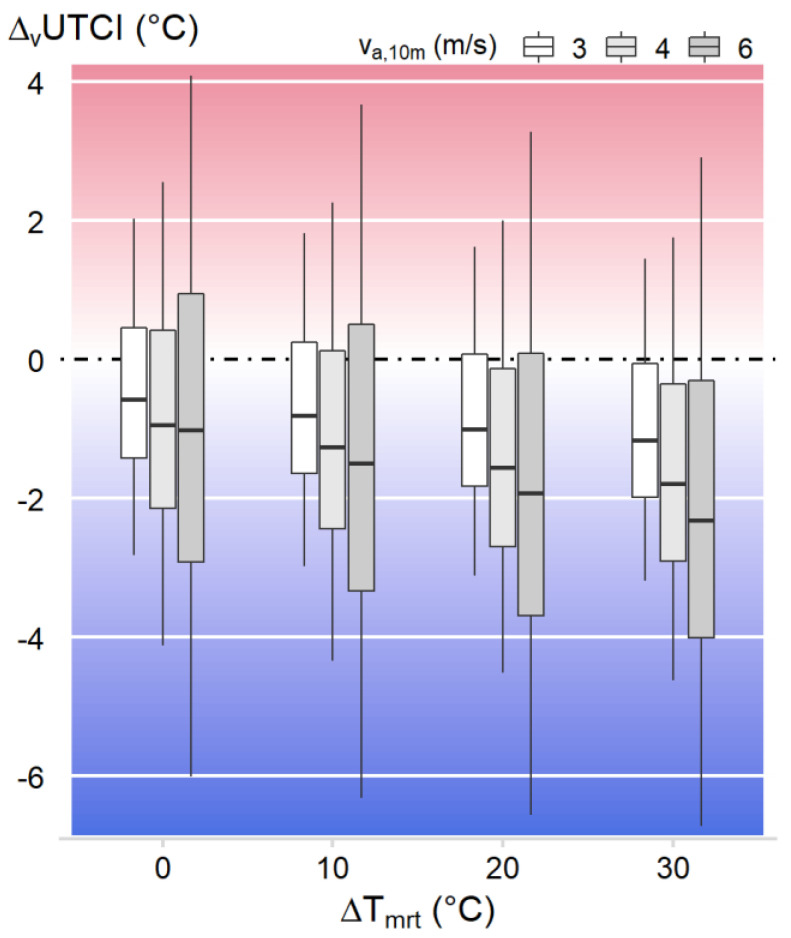
Boxplots including the percentiles P0 (minimum), P25 (1st quartile), P50 (median), P75 (3rd quartile), and P100 (maximum) summarizing the effect of three levels of increased wind speed (*v_a*,10*m_*) on *UTCI* (*Δ_v_UTCI*) over air temperatures between 25 and 50 °C and vapor pressure of 0.1−5 kPa in relation to radiant heat load (*ΔT_mrt_ = T_mrt_* − *T_a_*), cf. Figure A3.

**Table 1 biology-12-00802-t001:** Fitted generalized additive models (GAMs) predicting experimental data on heart rate (*HR*), rectal temperature (*T_re_*), skin temperature (*T_sk_*), and sweat rate (*SR*), respectively, depending on combinations of air temperature (*T_a_*) and vapor pressure (*p_a_*), considering wind condition (*RefWind*, *HiWind*) as modifier and including a random subject effect. Abbreviations for model parameters, as in Equation (2).

	*HR* (bpm)	*T_re_* (°C)	*T_sk_* (°C)	*SR* (g/h)
*Observations (#missing values)*	189 (9)	198 (0)	189 (9)	186 (13)
*Goodness-of-fit*				
Adjusted *R*^2^ (%)	78.8	76.2	88.6	89.9
Residual standard error	7.6	0.2	0.4	120.1
*Intercept µ*				
Mean estimate	102.5	37.6	35.5	744.4
SE	0.8	0.1	0.2	27.1
*p*-value	<0.0001	<0.0001	<0.0001	<0.0001
*s(ID)*				
edf	0.5	3.8	3.8	3.4
Ref.df	4.0	4.0	4.0	4.0
F-value	0.2	27.1	24.3	6.1
*p*-value	0.2018	<0.0001	<0.0001	<0.0001
*te(T_a_, p_a_)*				
edf	14.6	6.5	10.3	10.0
Ref.df	19.0	8.2	14.0	13.4
F-value	18.9	34.7	31.5	51.3
*p*-value	<0.0001	<0.0001	<0.0001	<0.0001
*te_Δv_(T_a_, p_a_)*				
edf	4.0	4.0	5.5	4.5
Ref.df	4.1	4.0	6.2	4.8
F-value	10.8	7.0	4.8	8.3
*p*-value	<0.0001	<0.0001	0.0001	<0.0001

Notes: SE: standard error; edf: effective degrees-of-freedom; Ref.df: reference degrees-of-freedom [55]; *s*(*ID*): random subject effect spline; *te(T_a_, p_a_)*: temperature–humidity effect (bivariate tensor product spline); *te_Δv_(T_a_, p_a_)*: temperature–humidity-dependent wind effect of *HiWind* compared to *RefWind* (bivariate interaction spline).

## Data Availability

The data presented in this study are available on request from the corresponding author. The data are not publicly available due to data privacy issues.

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
