# Peer review of "Temperature–Humidity-Dependent Wind Effects on Physiological Heat Strain of Moderately Exercising Individuals Reproduced by the Universal Thermal Climate Index (UTCI)"

_biology, 2023, doi:10.3390/biology12060802_

Round 1
Reviewer 1 Report
GENERAL:
The primary purpose of this study was to assess the impact of airflow on heat strain (HR, TC, TSK, SR) using a broader matrix of air temperature and moisture combinations, as well as solar heat load modeling estimates, than previously reported in the literature. Lesser studied moderate activity was also used. A second purpose was to assess performance of the UTCI heat strain model across the conditions tested. Both purposes have public health merit toward understanding when and how airflow is or is not beneficial for cooling humans. Some minor queries and comments follow for consideration.
SPECIFIC:
Introduction: Climate change and global warming are frequently positioned as reasons for conducting studies like this one. Average global temperatures will increase marginally but as a consequence bring with them more extreme weather events – including heat waves. Outdoors, airflow is equivalent to movement velocity (even with no wind), thus the introduction would be improved (and read more accurately) if the benefits or controversies over using airflow “indoors” were the focus. (Aside: I do appreciate that the authors modeled radiant heat differences since these ‘are’ an outdoor factor that many labs cannot simulate and ultimately ignore in the climate change story; see more below). Currently, lines 49-73 (and even 74-98) do not explain these important nuances of indoor vs outdoor, but instead focus on inactive vs active.
Methods & Results: In all the laboratory trials, Tg = Ta. It should be made clear that radiant heat load (Tmrt) effects are purely modeled outcomes using UTCI (e.g., lines 228-229; 266; Fig. 7; Fig. A3). Importantly, have the radiant heat load effects via UTCI been validated? If not, please add to discussion/limitations.
Lines 123-144: The explanations for airflow study are excellent, but could the authors elaborate on how they chose the airflow range? Is 2 m/s an upper range for typical commercial indoor convection? Is it for comparison to existing published works? Also – very minor – but the graphical abstract states 0.3 m/s for the low-end whereas everywhere else is stated 0.2 m/s.
Figure 1 and A1. Under hot, humid conditions (Figs. 1a, A1a), the authors explain the higher sweating rates with lower airflow essentially as the consequence of evaporative inefficiency, which most people understand intuitively. However, the higher sweating rates at higher airflow in very hot, dry conditions are not explained (apologies if I missed it). Can the authors please elaborate for readers whether this is explained by convective heat gain or something else?
Author Response
We like to thank this reviewer for the positive feedback concerning the general value of our paper and appreciate the constructive specific comments concerning the structure of the introduction, especially considering the purpose and aims of our analyses. We have considered these, as well as the further comments concerning methods and results as detailed below.
Concerning the introduction, we agree with this reviewer that the utility of electric fans and ventilators for heat stress mitigation is predominantly discussed for indoor or stationary conditions, though there are some facets concerning outdoor thermal stress, e.g. the ‘poison wind’ (simoom) mentioned in the introduction and discussion. In response to this comment, we have endeavoured to emphasize the ‘indoor vs outdoor’ aspect at several places in the introduction, as follows:
“Elevating air movements by outdoor wind or ventilators is a cost-effective and sustainable measure for mitigating physiological heat strain predominantly considered indoors and at stationary workplaces [5-7]. However, policies and public health authorities have advised against electric fan use under heat wave conditions indoors with air temperatures above the typical skin temperature of 35 °C, because convective cooling will then turn to convective heating of the body [8-10].”
“Notably, the simulations of heat wave scenarios focussed on indoor settings and only considered resting conditions concerning metabolic rates of about 1 MET (1 MET=58.2 W/m2), whereas physical load at workplaces and many home and outdoor activities are associated with metabolic rates higher than 2 MET [22,23].”
“Specifically, with a focus on moderately active persons, there is a lack of data from con-trolled laboratory experiments specifying wind effects on physiological heat strain in terms of heart rates, core and skin temperatures, and sweat rates while covering a wide grid of air temperature-humidity combinations, which determine the indoor thermal environment.”
Concerning radiant heat load considerations in the methods and results, we agree that the effects of heat radiation modelled by UTCI will require validation by targeted experimental studies, and we emphasize this in the discussion at the end of section 4.1, mentioning supporting results from thermal manikin measurements:
“Similar interaction effects with wind decreasing the heat gain from radiant heat had been reported from thermal manikin measurements with work clothes [59,60]. However, as our results concerning the effects of heat radiation are purely modeled outcomes using UTCI, further supporting experimental studies with human participants concerning the interaction of wind with radiant heat load are required.”
Concerning validation of UTCI and the underlying models with respect to radiant heat load (and other factors), we added more details to the corresponding text when introducing UTCI in section 1.1, which now reads:
“The model was extensively validated against laboratory data and field observations [35], and showed good agreement with thermal environment standards and experimental data in occupational settings [36-38] concerning the moderating influence of humidity, wind and radiant heat load in addition to air temperature on both cold and heat stress.”
Concerning the range of airflow levels in our study, 2 m/s was the maximum wind speed controllable in our climatic chamber, which had been constructed to allow the simulation of heat stress at work, where a value of 2 m/s is well above indoor conditions, but can be found at several (semi-)outdoor work places or at coal mines. We have added this information to section 2.1 as follows:
“We searched our database for series of heat stress experiments with participants exercising on a treadmill under both reference (RefWind, with air velocity va = 0.3 m/s) and high wind (HiWind, va = 2 m/s) conditions. Here, 2 m/s constituted the maximum value of controllable wind speed in the climatic chamber corresponding to average airflow conditions observable at (semi-) outdoor workplaces and in coal mines [24].”
The low-end reference wind speed was 0.3 m/s as stated in the abstract, and we checked the whole manuscript carefully to verify that this was stated correctly in the text, tables and figures.
Concerning the interpretation of Figures 1 & A1, we agree that increased convective heat gain by wind, which was not compensated by a further increase of the already high level of evaporative efficiency under hot-dry conditions, provides a probable explanation for the increased sweat rates at higher airflows under hot-dry conditions, following the reasoning from ref. [28]. Thus, we have added an elaborated statement at the end of the corresponding paragraph, reading as follows:
“On the other hand, under hot-dry conditions (Fig. 1b), the already high level of evaporative efficiency could not be further elevated by increased airflow, and thus could not compensate for the higher convective heat gain aggravating physiological strain, in particular rising sweat rates [28].”
Please also refer to the attachment and the tracked changes detailed in the revised manuscript.

Reviewer 2 Report
Overall this is a well-written study however, I have some feedback regarding stylistic reporting.
Introduction
While the figures are helpful in the introduction, I think they are more confusing than anything else. I would suggest simply focusing on what the literature states on heat, cooling fans and skin temp, heart rate, etc instead of presenting the figures. This is especially confusing because you state that there is a lack of evidence from a laboratory setting yet you provide evidence from a laboratory setting.
Methodology
Since you got this information from a database were these the same 5 subjects? If not, how many different subjects were there? And if these are just 5 subjects, can you please provide ranges and median instead of means and SD for their characteristics.
Since you report the UTCI results first in the results section, I would recommend doing the same in the statistical analysis section.
You bring this up later in the results, but why did you not use a data imputation technique to account for missing values?
Results:
The results are well written.
Discussion
While the discussion was well written you should try to have a take home message in the first paragraph of your discussion
Author Response
We thank this reviewer for the positive attitude concerning the value of our study, and we appreciate the feedback on style of reporting, which we have considered as detailed below.
We endeavored to enhance the clarity of the introduction as requested, and have moved the subsection on UTCI wind sensitivity including the equations and Figure 2 to the methods section referring to UTCI calculation.
On the other hand, we decided to keep Figure 1 in the introduction as motivating example, as it nicely illustrates that beneficial wind effects could occur at air temperatures above skin temperature in humid conditions (cf. corresponding comment of Reviewer 1). To avoid confusion, we now emphasize that this example is not sufficient to provide scientific evidence in the introduction:
“In spite of the aforementioned research, and although Figures 1 & A1 exhibit motivating examples, more systematic studies are required for quantification and evidence-based conclusions.”
Concerning methodology, and in response to the comment referring to our five study participants, we have now provided more details in the methods section about our heat strain database, comprising almost 2,400 climate chamber experiments with 33 young males:
“This experimental study is based on a database compiled from about 2,400 climate chamber experiments with 33 young male adults conducted previously at IfADo [24,44] under widely varying climatic conditions, clothing and activity levels.”
Indeed, the study sample consisted of five participants meeting the inclusion criteria as defined in the methods section. As the study population of young healthy males with normal physical fitness was quite homogeneous, the distribution of personal characteristics is adequately described by mean ± SD, however, in response to this request, we added the range (minimum and maximum values) in parentheses, as we had already done for the climatic data.
Concerning aligning the structure of the statistical analysis and results sections, respectively, we had considered this recommendation, but then decided to keep the current structure describing the experimental methods first, as this conforms to the experimental type of our study (cf. second remark of Reviewer 3). We start the results section with UTCI, because we think that it helps the reader to more easily apprehend the presentation of the data as contour plots accompanied by difference plots, when he/she has to consider one variable (UTCI) only in comparison to four variables representing measured physiological heat strain.
We did not opt for data imputation, because the percentage of missing physiological responses was rather low (between 0 and 6.6%), and because the diagnostic plots (Figure A.2) did not indicate any issues concerning the underlying model assumptions, so that the resulting estimates from the statistical model appear to be robust against this small number of missing observations.
We appreciate the positive feedback concerning the results section.
Following the hint on including a take home message in the discussion, we added the following text as first paragraph to the discussion:
“This study achieved its primary aim to provide heat stress thresholds over a wide matrix of temperature-humidity combinations separating cooling from heating wind effects based on experimental evidence for moderately exercising individuals.”
Please also refer to the attachment and the tracked changes detailed in the revised manuscript.

Reviewer 3 Report
The authors have submitted a manuscript aiming to, using previously recorded data, analyse heart rates, core and skin temperatures and sweat rates in 198 laboratory experiments under widely varying temperature-humidity combinations and two wind conditions, and assess the agreement between observed wind effects and the assessment performed by UTCI.
I congratulate the authors for their work, which is relevant and of interest to the readership of the journal. The manuscript is well written, the research is well framed within the introduction, data analysis is mostly very well conducted, the authors have provided an adequate discussion and provided an adequate conclusion.
I have only two minor and one moderate remark. As minor remarks, the aim of the study must be clarified as I believe it is not formulated clearly and, in the methodology, please start by classifying the study type.
As a moderate remark, the methodology to assess agreement must be clear and described in the data analysis section. It seems agreement was assessed through correlation analysis but assessing agreement through correlation analysis has its limitations and the authors should seek to improve the analysis or address the potential limitations of the conducted analysis.
Author Response
We are grateful to this reviewer for appreciating the value and content of our manuscript, and for providing hints for its improvement, which we followed as detailed below.
In response to the first minor remark concerning clarifying the aim of our study, we added a more specific description of our approach to the corresponding subsection 1.2, as follows:
“Our approach was to determine the transition from cooling to heating wind effects on heart rates, sweat rates, core and skin temperatures measured in experiments with exercising participants over a comprehensive matrix of temperature-humidity combinations, and to compare the outcomes to the UTCI assessment.”
Concerning the second minor remark on classifying the study type, we now start the Methods section stating the experimental nature of our study, as requested:
“This experimental study is based on a database compiled from…”
In response to the comment concerning the methodology for assessing agreement, we performed additional analyses extending the qualitative description in section 3.3 by quantifying the predictive performance of UTCI concerning heating wind effects on physiological variables by calculating sensitivity, specificity, and accuracy, and visualized the results in the new Figure 6b.
We added a corresponding sub-sub-section 2.3.3 to the methods:
“2.3.3. UTCI prediction of physiological wind effects
We compared the UTCI assessment of the wind effects ΔvUTCI with the wind effects obtained for the physiological variables by scatterplots including Spearman correlation coefficients (rs). In addition, we quantified the predictive performance of UTCI in the binary classification of the occasion of heating wind effects (Δv>0) on physiological variables by calculating the sensitivity, specificity, and overall accuracy.”
The corresponding paragraphs in the results section 3.3 now read:
“In addition to correlation analyses, the plots give an indication for the capacity of UTCI for correctly classifying conditions with physiological wind cooling or heating [56]. The dashed reference lines in Figure 6a divide the plot in quadrants with the lower left quadrant showing the physiological cooling events correctly assessed by UTCI and the upper right containing the correctly assessed heating events, which dominate concerning Tsk and SR.
Whereas hardly any data points were present in the upper left quadrant with wind cooling falsely predicted by UTCI, the conditions collected in the lower right quadrants (falsely predicted heating) concur with a more conservative UTCI assessment of the transition from cooling to heating wind concerning HR and Tre.
When quantifying the UTCI performance in predicting the occasion of heating wind effects (Δv>0) as sensitivity, specificity, and accuracy in Figure 6b, these falsely predicted heating events resulted in lowered specificity and overall accuracy regarding HR and Tre, whereas well-balanced figures emerged for Tsk and SR.”
Please also refer to the attachment and the tracked changes detailed in the revised manuscript.

Round 2
Reviewer 2 Report
I'd like to thank the authors for addressing my comments. One minor comment, I believe that the authors mis-understood me when I was talking about changing the statistical analysis section.
To make it easier for the reader, can you please write your statistical analysis section in the order in which you present your results.
Author Response
We are grateful to this reviewer for largely accepting our reply to the previous comments. Actually, we had already seriously considered and discussed the specific comment on aligning the order of topics presented in the methods and results sections in the previous revision round.
We strongly believe that the experimental data have to be described first in the methods, because they determine the experimental nature of our study (cf. previous comment by Reviewer 3 concerning type of study), and because the climatic conditions for UTCI calculations had been primarily chosen to match the experimental conditions.
On the other hand, we think that it will help the reader to more easily apprehend the structure of our analysis approach presenting the results as contour plots accompanied by difference plots, when he/she has to consider one variable (UTCI) only in comparison to four variables representing measured physiological heat strain.
Thus, we are still convinced that there are good reasons for keeping the slight deviation in the order of topics in the results. However, as we appreciate the concerns of this reviewer on readability, we have extended the first paragraph of the results in an attempt to clarifying the changed order and explaining the reasons for doing so. The results section now starts as:
“In order to illustrate the structure of our data analysis approach, and deviating from the order of topics in the methods section with the experimental study preceding UTCI calculations, we first present in section 3.1 the wind effects (HiWind vs RefWind) evaluated by UTCI. Section 3.2 contains the corresponding analyses for the experimental heat strain data, while section 3.3 relates the physiological wind effects to the UTCI assessment. Eventually, section 3.4 shows the results for the supplemental UTCI analyses considering higher wind speeds and the influence of thermal radiation.”
We hope that you will consider this an acceptable compromise for addressing your remaining concerns on readability.
